# Impact of Air Pollution on Sedentary Behavior: A Cohort Study of Freshmen at a University in Beijing, China

**DOI:** 10.3390/ijerph15122811

**Published:** 2018-12-10

**Authors:** Hongjun Yu, Jiali Cheng, Shelby Paige Gordon, Ruopeng An, Miao Yu, Xiaodan Chen, Qingli Yue, Jun Qiu

**Affiliations:** 1Department of Physical Education, Tsinghua University, Beijing 100084, China; chengjiali@mail.bnu.edu.cn (J.C.); qiujun@tsinghua.edu.cn (J.Q.); 2Department of Interdisciplinary Health Sciences, University of Illinois at Urbana-Champaign, Champaign, IL 61820, USA; spgordo2@gmail.com; 3Department of Kinesiology and Community Health, University of Illinois at Urbana-Champaign, Champaign, IL 61820, USA; ran5@illinois.edu; 4Renmin University of China Libraries, Beijing 100872, China; yumiao@ruc.edu.cn; 5Department of Martial Art, Guangzhou Sport University, Guangzhou 510500, China; cxdgt@163.com; 6Department of Olympic Games, Beijing Sport University, Beijing 100084, China; yueqing0176@sina.com

**Keywords:** air pollution, AQI, fine particulate matter, sedentary behavior, youth

## Abstract

Human populations worldwide have experienced substantial environmental issues in part due to air pollution, notably in China. Gaps in the scientific literature remain regarding the relationship between air pollution and sedentary behavior among young adults in China. The purpose of this study is to examine the effect of air pollution on sedentary behavior among college students living in Beijing, China. We conducted follow-up health surveys on 12,174 freshman students enrolled at Tsinghua University from 2013 to 2017. Sedentary behavior was measured using the short version of the International Physical Activity Questionnaire (IPAQ). Corresponding air pollution data measured by the Ministry of Environmental Protection of the People’s Republic of China were collected to include the average hourly air quality index (AQI), PM_2.5_, PM_10_, and NO_2_ (µg/m³). The data were analyzed using linear individual fixed-effect regressions. An increase in air pollution concentration of one standard deviation in AQI, PM_2.5_, PM_10_, and NO_2_ was associated with an increase in weekly total hours of sedentary behavior by 7.35 (95% confidence interval (CI) = 5.89, 8.80), 6.24 (95% CI = 5.00, 7.49), 6.80 (95% CI = 5.46, 8.15), and 7.06 (95% CI = 5.65, 8.47), respectively. In the presence of air pollution, women students tended to increase their sedentary behavior more than men. Air pollution increases sedentary behavior among freshman students living in Beijing, China. Replication of this study is warranted among various populations within China.

## 1. Introduction

Exposure to air pollution in countries around the world has become a major health concern. Repeated evidence indicates that there are detrimental effects of air pollution on human health and health-related behavior. The World Health Organization reported that approximately 4.2 million deaths occurred due to ambient air pollution in 2015 [1]. In recent decades, China has experienced rapid industrialization and urbanization, which has contributed to increased levels of air pollution. Air pollution is the combination of gases (NO_2_) with particulate matter particles of solid and/or liquid elements with a diameter of less than 2.5 µm (PM_2.5_) or 10 µm (PM_10_) [2]. Particulate matter components and gases interact with the ozone layer following sunlight exposure, forming an irritating photochemical cocktail known as smog [2]. In several studies, exposure to air pollution has been associated with various adverse health outcomes, e.g., cardiovascular disease, myocardial infarction, stroke, lung cancer, respiratory disease, and all-cause mortality [3,4,5,6]. In one United States (U.S.) study, Di et al. found that long-term exposure to PM_2.5_ and ozone was associated with increased all-cause mortality from 2000 to 2012 [7]. Air pollution in China has been associated with risk for respiratory disease [2], lung cancer [8], and premature death [9]. A recent study estimated that air pollution in North China reduced life expectancy by 3.1 years due to cardiorespiratory mortality [10]. Air pollution in China was the fourth leading cause of premature mortality, and was responsible for 1.2 million deaths in 2010, which is almost 40% of the global total [11].

Significant research has been done regarding the health benefits of regular physical activity (PA). Health benefits include a reduced risk of all-cause mortality, increased effective weight management, and the prevention and management of chronic diseases, e.g., cardiovascular diseases, diabetes, colon and breast cancer, hypertension, coronary heart disease, osteoarthritis, and depression [12]. However, previous research has reported that PA engagement under high levels of air pollution can increase the risk of health problems, ranging from asthma attacks to heart or lung pathologies [13]. PA and exercise among Chinese adolescents and adults are commonly performed in outdoor settings such as playgrounds and parks [14]. It is estimated that a vast majority of Chinese adolescents and 77% of Chinese adults fall short of the recommended guidelines for moderate to vigorous physical activity (≥60 min per week) [15]. Physical inactivity is not limited to China. A significant proportion of people worldwide are physically inactive.

Sedentary behavior (SB) is defined as any behavior that maintains a low-energy expenditure (≤1.5 metabolic equivalents (METs)), such as sitting or reclining [16]. An international survey that collected data from 20 countries found that youth and adults spend on average six to eight hours per day being sedentary [17]. Public awareness of the negative impacts of air pollution (e.g., media alerts on smog levels) may be responsible for the reduction in PA. Prolonged sedentary activities such as screening (i.e., watching television, playing video games, and surfing the Internet), listening to music, and reading may be due to an increase in public awareness [18,19]. Independent of PA level, spending excess time in SB has been found to result in adverse health outcomes, including increased risk of all-cause mortality, metabolic syndrome, and obesity among adolescents and adults [16,20,21,22,23].

A considerable amount of research has associated air pollution with adverse health risks [24,25], but much less is known regarding its impact on health behaviors, notably PA and SB [26]. Currently, there are few cohort-based studies that have examined the influence of PM_2.5_ on PA among college students and university graduates in China [27,28]. A recent systematic review has linked air pollution to a decrease in PA [26]. Among the studies included in the review, six were conducted in the United States (US) [18,19,29,30,31,32], one was conducted in the United Kingdom (UK) [33], and one was conducted in Mexico [34]. In addition, Yu et al. found an inverse relationship between high PM_2.5_ concentration and daily PA among college students and retired older adults in China [27,28].

The few numbers of studies regarding the impact of PA and SB indicate two significant gaps in the current scientific literature that warrant further investigation.

First, previous studies have exclusively focused on the effect of air pollution on PA and physical inactivity, but to our knowledge, no research to date has examined the relationship between air pollution and SB. Second, most of the existing studies have not assessed behavioral changes in response to temporal variations in air pollution. Cross-sectional data that is subject to confounding bias due to unobserved differences in individual characteristics have led to data unavailability.

This study examined the longitudinal relationship between air pollution and SB among university freshmen in Beijing, China over a four-year study period from 2013 to 2017. The purpose of the present study is to investigate the impacts of air pollution on SB among college students living in Beijing, China. We hypothesized that in response to elevated air pollution, college students might increase their SB.

## 2. Materials and Methods

### 2.1. Participants and Sampling Procedure

A paper–pencil-based health survey was conducted on a regular basis during students’ freshman year at Tsinghua University in Beijing, China. Study participants completed the survey during a required university class for all freshmen. It included questions regarding one’s sociodemographic and physical and mental health conditions. Survey participation was voluntary. Upon signing the consent form, survey participants were asked to complete a paper–pencil-based questionnaire and hand it in at the end of the class period. Faculty administered the survey in a class, and all of the freshmen completed the survey within a one-week health education class window (Monday to Friday). The same survey was administered as a follow-up twice to the 2013–2014 cohort (16 to 20 December 2013 and 12 to 16 May 2014); three times to the 2014–2015 cohort (13 to 17 October 2014; 2 to 6 March 2015; and 11 to 15 May 2015); two times to the 2015–2016 cohort (21 to 25 September 2015 and 9 to 13 May 2016); and two times to the 2016–2017 cohort (28 November to 2 December 2016 and 22–26 May 2017). This study used the panel survey data from these four freshman cohorts. The policy and circumstance of the university did not change during the four cohort survey periods. Survey participants were asked to report their student identification number, which was then used to link multiple survey questionnaires completed by the same respondent.

The total population of freshmen at Tsinghua University (13,500 students) was given the survey. A total of 12,174 participants (90% of the total population) completed the survey. In the 2013–2014 cohort, 3300 participants were given a survey, and of that number, 3025 (92%) responded to the survey. In the 2014–2015 cohort, 3400 participants were given a survey, of which 3019 (89%) responded to the survey. In the 2015–2016 cohort, 3300 participants were given a survey, and 2936 (89%) responded to the survey. Finally, 3500 participants were given a survey in the 2016–2017 cohort, and of that number, 3194 (91%) responded to the survey. Data were restricted to the participants who completed the survey at least twice. Participants who completed a survey only once (*n* = 1791) were excluded from the analyses. Thus, a total of 638 individuals with missing data on individual characteristics (i.e., sex, race, body height/weight, smoking status, and general health status) were excluded. The final sample comprised 9700 participants (see Figure 1).

All subjects gave their informed consent before they participated in the study. The study was conducted in accordance with the Declaration of Helsinki, and protocol was approved by the Tsinghua University Institutional Review Board (IRB #2012534001).

### 2.2. Measures

#### 2.2.1. SB Measurement

The short version of the International Physical Activity Questionnaire (IPAQ), which has been validated in China, was used to measure SB [35].

Total hours of SB in the last week were constructed based on the answer to one question from the IPAQ: “During the last seven days, how much time did you usually spend sitting on one of those days”? [36]. Total hours of SB in the last seven days were calculated by multiplying average daily hours spent engaging in SB by seven.

#### 2.2.2. Environmental Measures

Environmental measures included average air quality index (AQI), PM_2.5_, PM_10_, NO_2_ (µg/m³), average daytime temperature (°C), average wind speed (m/s), and percentage of rainy days in Beijing, China over the last seven days. Hourly air pollution concentration data came from the Ministry of Environmental Protection of the People’s Republic of China, and daily weather data, including daytime temperature, wind speed, and percentage of rainy days, came from the China Meteorological Administration.

To facilitate the interpretation of results, we standardized average air pollution concentration over the last seven days through centering (i.e., subtracting the mean from each value), and then dividing by its standard deviation (SD) (i.e., AQI *z*-scores). Thus, the estimated coefficient of air pollution concentration can be interpreted as the change in outcome variable with respect to the change in air pollution concentration by one SD. Substantial variations in air pollution concentration were present from 2013 to 2017, with mean concentrations (±SD) of AQI, PM_2.5_, PM_10_, and NO_2_ in 119.41 ± 91.45 µg/m³, 81.16 ± 83.75 µg/m³, 111.47 ± 85.77 µg/m³, and 48.60 ± 25.98 µg/m³, respectively.

The basic *z*-score formula for a sample is *z* = (*x* − *μ*)/*σ*. For example, the first survey that was given to the 2014–2015 cohort produced AQI *z*-scores of *z* = (*x* − *μ*)/*σ* = 218.14 − 119.41/91.45 = 1.08.

### 2.3. Statistical Methods

Descriptive statistics, including the mean, SD, and percentages were used to summarize and compare the characteristics of the overall sample. Chi-square tests were conducted to compare categorical variables. ANOVA tests and t-tests were used to compare for continuous variables. One-way repeated measures ANOVA tests were conducted to compare the differences in SB between the cohort follow-up surveys. Linear individual fixed-effect regressions were performed based on the repeated-measure survey data from the four freshman cohorts: 2013–2014, 2014–2015, 2015–2016, and 2016–2017. The continuous outcome variable was the total hours of SB within the last seven days prior to the survey. The key independent variables were AQI, PM_2.5_, PM_10_, and NO_2_ z-scores during this time. Individual-level time-variant covariates and environmental measures, including average daytime temperature, average wind speed, and percentage of rainy days over the last seven days, were controlled for the aforementioned. Separate regressions were conducted for each outcome variable and based on samples stratified by gender; i.e., separate regressions were based on the entire sample with both genders, as well as male only and female only.

Compared to the conventional pooled cross-sectional regression, individual fixed-effect regression is preferred. Its sole use of within-individual variations in PA level to identify the impacts of air pollution concentration removes any potential omitted variable bias due to differences in time-invariant individual characteristics such as ethnicity, habits, and personal preferences. Due to the exclusive dependence on within-individual variations in an outcome measure, individual fixed-effect regressions could have only estimated the effect of a time-variant independent variable. Therefore, time-invariant individual characteristics such as ethnicity were not examined.

All of the statistical procedures were performed in Stata 14.2 SE version (StataCorp, College Station, TX, USA). The Eicker–Huber–White sandwich estimator addressed within-individual serial correlations, and was used to estimate the standard regression coefficient errors.

The following individual-level time-variant covariates were controlled for in the regression analyses: a continuous variable for age in years, a continuous variable for body mass index (BMI; kg/m^2^) calculated from self-reported height and weight, a dichotomous variable for current smoking status (current non-smokers as the reference group), a dichotomous variable for current drinking status (current non-drinkers as the reference group), a continuous variable for self-rated physical health (1–10, poor to excellent), and a continuous variable for self-rated mental health (1–10, poor to excellent).

## 3. Results

### 3.1. Descriptive Statistics

Regarding the selection of survey participants, Figure 1 presents the analytic sample selection flowchart for the follow-up surveys administered by cohort. The sample size in 2013–2014, 2014–2015, 2015–2016, and 2016–2017 was 3025 (66.5% males), 3019 (65.1% males), 2936 (68.1% males), and 3194 (68.3% males), respectively, resulting in a total sample size of 12,174. All of the freshmen in the total sample took the survey at least twice. Among the sample, 9700 freshmen (dependent upon specific outcome variables) had non-missing values for the specific outcome and all of the covariates, and thus were included in the analyses.

Regarding the characteristics of the survey participants, Table 1 summarizes the baseline characteristics. The distribution of total sample size by cohort in 2013–2014, 2014–2015, 2015–2016, and 2016–2017 was 24.85%, 24.80%, 24.12%, and 26.24%, respectively. The mean BMI was 21.33 kg/m^2^ (SD = 3.36), and the mean age was 18.11 years (SD = 0.75). Females accounted for only one third (32.98%) of the study sample, and a small proportion of the total sample size was current smokers (0.42%) and drinkers (2.59%). Self-rated physical health and self-rated mental health scores averaged 5.58 (SD = 1.77) and 6.63 (SD = 1.94), respectively. Survey participants on average engaged in 9.23 h of daily SB (SD = 2.82) in the last week.

### 3.2. The Relationship between Air Pollution and SB

Table 2 reports the average amount of time spent in SB by study cohort, air pollution concentration, and other environmental variables in Beijing, China over the last seven days before the survey. Based on repeated measures ANOVA analysis, AQI decreased from 218.14 (SD = 141.55) to 68.43 (SD = 16.57) (*p* < 0.001), and weekly total hours of SB decreased from 64.99 h/week (SD = 18.96) to 55.43 h/week (SD = 19.99) (*p* < 0.001) within the 2014–2015 cohort.

Table 3 reports the estimated effects of air pollution on student SB by gender using linear individual fixed-effect regressions. A one SD (i.e., 119.41) increase in AQI concentration was associated with an increase in total weekly hours of SB by 7.35 (95% confidence interval (CI) = 5.89, 8.80).

An increase in particulate matter concentration in PM_2.5_ and PM_10_ by one SD (i.e., PM_2.5_ in 81.16 µg/m³ and PM_10_ in 111.47 µg/m³) was associated with an increase in total weekly hours of SB by 6.24 (95% CI = 5.00, 7.49) and 6.80 (95% CI = 5.46, 8.15), respectively.

An increase in gaseous pollutants matter concentration in NO_2_ by one SD (48.60 µg/m³) was associated with an increase in total weekly hours of SB by 7.06 (95% CI = 5.65, 8.47).

Women tended to increase their time spent in SB in response to air pollution more than men. A one SD increase in AQI was associated with an increase in the total weekly hours of SB among females and males of 8.34 (95% CI = 5.62, 11.60) and 6.76 (95% CI = 5.03, 8.48), respectively. A one SD increase in PM_2.5_ concentration was associated with an increase in the total weekly hours of SB among females and males of 7.18 (95% CI = 4.84, 9.53) and 5.70 (95% CI = 4.23, 7.18), respectively. A one SD increase in PM_10_ concentration was associated with an increase in the total weekly hours of SB among females and males of 7.72 (95% CI = 5.20, 10.24) and 6.25 (95% CI = 4.66, 7.85), respectively. A one SD increase in NO_2_ concentration was associated with an increase in the total weekly hours of SB among females and males of 8.17 (95% CI = 5.51, 10.84) and 6.43 (95% CI = 4.76, 8.10), respectively.

Regarding other environmental measures and individual-level covariates (results not shown in the table), a positive association among SB, wind speed, and percentage of rainy days over the last seven days was consistent, whereas the average daytime temperature was inversely associated with SB. Both high self-rated physical health and high self-rated mental health were negatively associated with SB in the past week.

## 4. Discussion

The purpose of this study was to analyze the impact of air pollution on human health-related behavior, specifically the association between air pollution and SB among freshman students enrolled at a university in Beijing, China. Our findings indicate that rises in air pollution are consistent with increased time spent in sedentary activities. A one SD increase in AQI, PM_2.5_, PM_10_, and NO_2_ was associated with an increase in total weekly hours of SB by 7.35, 6.24, 6.80, and 7.06, respectively. In the presence of air pollution, women tended to increase their SB more than men.

Our findings on the positive relationship between air pollution and SB are in line with those of previous studies that have reported air pollution to be positively associated with physical inactivity [18,19,29]. Previous work done by An et al. linked PM_2.5_ to increased odds of physical inactivity among US adults [29]. One other study in the US found high levels of PM_10_ to be correlated with reduced levels of PA [18]. However, these prior studies were all based on cross-sectional survey data on US adult populations [18,19,29]. The present study used data from a cohort of Chinese youth populations.

The findings of this study are also consistent with earlier research, including those of our work [27,28] and of one Chinese study that reported that the increased levels of air pollution associated with decreased PA and PM_2.5_ were positively linked to increased SB [37]. This study coincides with our previous work, which showed a one SD increase in PM_2.5_ concentration with a reduction in weekly moderate to vigorous physical activity by 97.5 min [27] and reduced leisure-time PA scores by 110.7 min among Chinese older adults [28]. In addition, we found that a one SD increase in AQI was associated with an increase in total weekly hours of SB by 7.35. High levels of SB have been recognized as a major health risk for metabolic syndrome, type 2 diabetes, and cardiovascular disease [21,22]. Severe air pollution levels may not only impact people’s health outcomes; it might also increase their SB. High levels of air pollution and SB have become serious threats to public health in China. Furthermore, this study is consistent with a previous study that indicated that a high AQI discouraged Chinese college students from participating in outdoor PA [37]. In contrast, one study reported that air pollution was negatively associated with television use among residents in Shanghai, China [38]. A possible explanation for this difference could be that television watching is a popular pastime among university-aged students. Another study examined the association between seasonal variations and PA using a sample of 40 participants, and reported no impact of ambient PM_2.5_ on PA [39]. However, that null finding could be due to the small sample size and lack statistical power [40,41].

Subgroup analyses reported that women increased their SB in the presence of air pollution more than men. Compared to males, females perceived poor air quality as a higher health risk, and were more likely to reduce their exposure by decreasing their outdoor activity. However, our findings on the gender differences are somewhat preliminary and warrant replication by future studies.

Well-supported evidence has indicated several health benefits of PA, but PA under high levels of air pollution may be detrimental to one’s health. Moreover, PA has become a significant public health concern in China [42]. While PA was found to have a protective effect against air pollution among older adults in Denmark [43], there is still evidence lacking regarding recommend outdoor PA under severe air pollution [44]. Regardless of an individual’s PA level, SB has been positively associated with obesity [45]. The prevalence of overweight and obesity is growing rapidly worldwide, notably in China [46,47,48]. Policy interventions that effectively reduce the air pollution levels in China may additionally decrease SB. However, it may take years before air pollution in China can be effectively restrained [14].

The strengths of this study reside in its longitudinal study design, its reliable and time-sensitive environmental measures, and its large cohort sample of freshmen at Tsinghua University from 2013 to 2017. However, there are a few major limitations to this study that should be noted. Analyzing freshman cohorts from a single university in Beijing, China is unlikely to represent the entire undergraduate population in the surrounding area or nationwide, limiting the generalizability of the study findings. Replicating future longitudinal studies with a representative sample can further produce generalizable estimates. Beijing spans across a large metropolitan area, and different neighborhoods might have different levels of air pollution concentration. Using the city-average air pollution measure could have masked considerable local variations. This generalization of exposures across individuals based on average air pollution is likely to undermine the individual variation in the uptake of pollution.

SB behavior was self-reported and subject to recall error and social desirability bias [49]. Participants may have reported fewer hours in sedentary behavior. It is recommended that future research use more objective ways to estimate sedentary behavior (e.g., accelerometers or activPAL). The individual fixed-effect models eliminated the confounding bias from factors that remained constant over time within participants, but they could not control for more transient, unobserved factors such as daily variations in pains and emotions. In addition, the impact of air pollution on the individual may depend on an individual’s socioeconomic status, co-morbidity, and housing, occupation, among many more factors.

## 5. Conclusions

The longitudinal relationship between air pollution and behavioral modification on weekly SB among university freshmen in Beijing, China was examined in this study. A positive association between AQI, PM_2.5_, PM_10_, NO_2_, and total weekly hours of SB among study participants was found. Public policy action might be urgently needed to reduce levels of air pollution in China. Future studies need to consider the replication of the study findings in other cities and universities.

## Figures and Tables

**Figure 1 ijerph-15-02811-f001:**
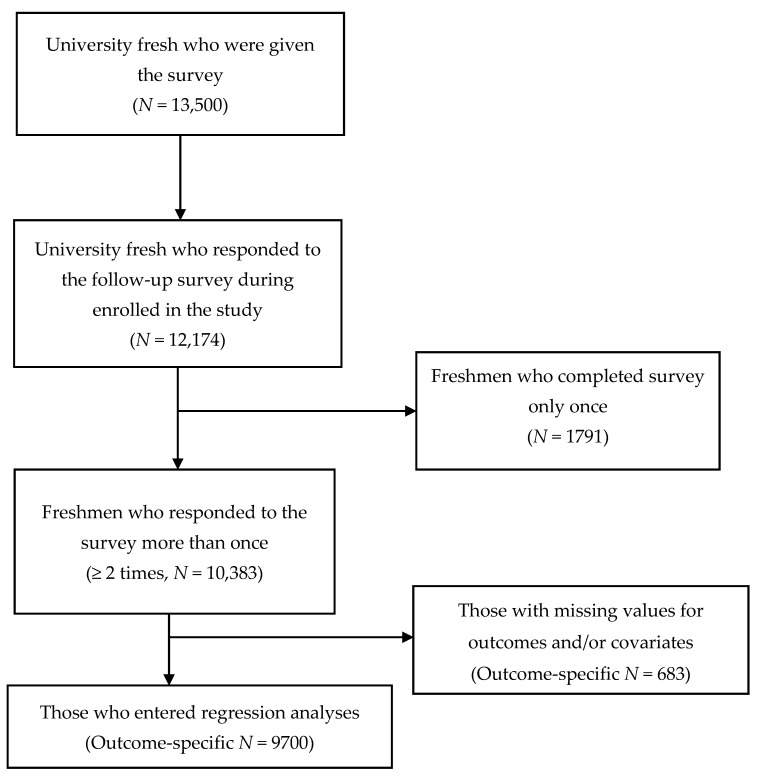
Study Sample Flowchart.

**Table 1 ijerph-15-02811-t001:** Baseline characteristics of survey participants.

Characteristics	Male	Female	Total	*p*-Value
Sex, *N* (%)	8159 (67.02%)	4015 (32.98%)	12,174	
Age (year), mean (SD)	18.12 ± 0.78	18.08 ± 0.68	18.11 ± 0.75	0.002
Freshman cohort, *N* (%)				
2013–2014	2010 (24.64%)	1015 (25.28%)	3025 (24.85%)	0.023
2014–2015	1966 (24.10%)	1053 (26.23%)	3019 (24.80%)	
2015–2016	2000 (24.51%)	936 (23.31%)	2936 (24.12%)	
2016–2017	2183 (26.76%)	1011 (25.18%)	3194 (26.24%)	
Body mass index, mean (SD)				
BMI (kg/m^2^)	21.89 ± 3.41	20.18 ± 2.92	21.33 ± 3.36	0.001
Smoking, *N* (%)	44 (0.54%)	7 (0.17%)	51 (0.42%)	0.003
Drinking, *N* (%)	260 (3.19%)	41 (1.02%)	301 (2.59%)	0.001
Self-rated physical health, mean (SD)				
Physical health score (1–10)	5.59 ± 1.81	5.56 ± 1.68	5.58 ± 1.77	0.215
Self-rated mental health, mean (SD)				
Mental health score (1–10)	6.64 ± 1.95	6.62 ± 1.92	6.63 ± 1.94	0.310
Disease number, mean (SD)	1.38 ± 0.87	1.42 ± 0.86	1.39 ± 0.87	0.979
SB (h/week), mean (SD) ^a^	9.20 ± 2.80	9.30 ± 2.86	9.23 ± 2.82	0.963

^a^ SB = Sedentary behavior, BMI = body mass index.

**Table 2 ijerph-15-02811-t002:** Average SB, air pollution concentrations, and other environmental variables in the last seven days before the survey.

Freshman Cohort	2013–2014 Cohort ^a^				2014–2015 Cohort ^b^						2015–2016 Cohort ^c^				2016–2017 Cohort ^d^			
Survey Order	First		Second		First		Second		Third		First		Second		First		Second	
*Dependent variables*																		
SB	65.03	(20.23) ***	65.23	(19.91) ***	64.99	(18.96) ***	62.99	(19.47) ***	55.43	(19.99) ***	62.39	(18.95) ***	60.79	(21.66) ***	65.75	(19.98) ***	64.43	(20.33) ***
*Air pollution measures*																		
AQI	53.29	(24.15)	79.14	(23.92)	218.14	(141.55) *	100.43	(61.01)	68.43	(16.57)	121.00	(52.83)	85.57	(20.15)	114.43	(109.55)	151.29	(44.96)
PM_2.5_ (µg/m³)	28.75	(15.41)	52.80	(21.63)	178.71	(128.66) *	69.59	(51.96)	36.06	(11.76)	89.24	(43.36) *	43.43	(17.75)	84.43	(96.55)	57.71	(21.04)
PM_10_ (µg/m³)	40.83	(16.47)	85.76	(36.61)	182.84	(128.69) *	97.81	(62.46)	79.93	(34.05)	104.16	(46.56)	103.57	(48.42)	118.29	(125.32)	114.00	(32.21)
NO_2_ (µg/m³)	34.48	(10.40)	45.73	(10.56)	70.77	(28.60) **	37.51	(12.73)	33.20	(8.63)	55.57	(7.93) **	32.43	(13.39)	57.14	(38.85)	35.71	(8.67)
*Environmental covariates*																		
Temperature (°C)	5.43	(1.40)	20.71	(3.09)	20.86	(1.86)	6.29	(3.45)	21.43	(5.16)	27.00	(0.82)	25.57	(2.44)	(5.57)	(1.72)	20.86	(5.18) ***
Wind (m/s)	3.57	(0.67)	3.14	(0.24)	3.43	(0.73)	3.79	(0.91)	3.01	(0.19)	3.00	0.00	3.14	(0.24)	3.29	(0.06)	3.15	(0.24)
Rain (%)	0.00	(0.00)	0.43	(0.54)	0.14	(0.38)	0.14	(0.38)	0.57	(0.54)	0.14	(0.38)	0.29	(0.49)	0.00	(1.00)	0.14	(0.38)

Notes: The results of SB using repeated measures one-way ANOVA analysis. Collection period of air pollution data. ^a^ 2013–2014 cohort: first (9–15 December 2013) and second (5–11 May 2014); ^b^ 2014–2015 cohort: first (6–12 October 2014), second (24 February–2 March 2015), and third (May 4–10, 2015); ^c^ 2015–2016 cohort: first (14–20 September 2015) and second (2–8 May 2016). ^d^ 2016–2017 cohort: first (21–27 November 2016) and second (15–21 May 2017) * *p* < 0.05. ** *p* < 0.01; *** *p* < 0.001. AQI: air quality index.

**Table 3 ijerph-15-02811-t003:** Estimated effects of air pollution on individual-level sedentary behavior outcomes by gender.

Dependent Variable	Male Only		Female Only		Total	
	Coefficient (95% CI)	# Observations (# Participants)	Coefficient (95% CI)	# Observations (# Participants)	Coefficient (95% CI)	# Observations (# Participants)
AQI						
Sedentary behavior in last week (h/week)	6.76 *** (5.03, 8.48)	12,605 (6675)	8.34 *** (5.62, 11.06)	5912 (3065)	7.35 *** (5.89, 8.80)	18,517 (9700)
PM_2.5_						
Sedentary behavior in last week (h/week)	5.7 *** (4.23, 7.18)	12,605 (6675)	7.18 *** (4.84, 9.53)	5912 (3065)	6.24 *** (5.00, 7.49)	18,517 (9700)
PM_10_						
Sedentary behavior in last week (h/week)	6.25 *** (4.66, 7.85)	12,605 (6675)	7.72 *** (5.20, 10.24)	5912 (3065)	6.8 *** (5.46, 8.15)	18,517 (9700)
NO_2_						
Sedentary behavior in last week (h/week)	6.43 *** (4.76, 8.10)	12,605 (6675)	8.17 *** (5.51, 10.84)	5912 (3065)	7.06 *** (5.65, 8.47)	18,517 (9700)

Notes: Separate individual fixed-effect regressions were performed to estimate the effects of air pollution concentrations on samples stratified by sex. Models were adjusted for all of the time-variant individual characteristics listed in Table 1 (i.e., age, BMI, smoking status, drinking status, self-rated physical health, and self-rated mental health) and the environmental variables listed in Table 2 (average daily temperature, average wind speed, and percentage of rainy days in the last week). *** *p* < 0.001. AQI = air quality index. #: the number in the regression.

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
