# Peer review of "Impact of Air Pollution on Sedentary Behavior: A Cohort Study of Freshmen at a University in Beijing, China"

_ijerph, 2018, doi:10.3390/ijerph15122811_

Reviewer 1 Report

First of all, I want to thank you for the opportunity to review this work.
I think it is a work that is well thought out, although I see some possible improvements to introduce:
In the Materials and Methods section, I think we should refer the total number of first-year students entering the University and how the participants have been recruited, if they are all and those who answer have been chosen, if they have been random. , consecutive, etc. And therefore on what total number is being studied. This point should be stated in Figure 1.
On the other hand, table 1 is cconfuse, you should put mean +/- SD and n (%). In addition, the percentage should be made on the top total, for example smokers would be 100 (44/8159) = 0.54% in male and in female 100 (7/4015) = 0.17%, which clearly shows the difference of smokers p = 0.003 in favor of males. The same for the other qualitative variables.

Author Response

We want to thank you for your helpful comments, which have helped significantly improve the manuscript. We have summarized our point-by-point responses/revisions to the comments and revisions made based on your helpful comments.

Reviewer’s comment 1:

First of all, I want to thank you for the opportunity to review this work. I think it is a work that is well thought out, although I see some possible improvements to introduce, In the Materials and Methods section.

Response: We are glad that that Reviewer 1 found this manuscript “I think it is a work that is well thought out”. We have significantly re-written the manuscript as suggested, we believe the introduction section, the material, and methods section of this manuscript is better now. We also want to thank Reviewer 1 for providing the detailed and helpful comments/suggestions for the manuscript.

Reviewer’s comment: 2. I think we should refer the total number of first-year students entering the University and how the participants have been recruited, if they are all and those who answer have been chosen, if they have been random. , consecutive, etc. And therefore on what total number is being studied. This point should be stated in Figure 1.

Response: 13,500 participants (total freshmen in the university) in total were given the survey. All freshmen got the questionnaire when she or he attend the class. The survey was administered during a required class for all freshmen. Survey participation was voluntary. 12,174 (90%) completed the survey. 3,300 participants were given a survey in the 2013-2014 cohort and of that number 3,025 (92%) responded to the survey, 3,400 participants were given a survey in the 2014-2015 cohort and of that number 3,019 (89%) responded to the survey, 3,300 participants were given a survey in the 2015-2016 cohort and of that number 2,936 (89%) responded to the survey and 3,500 participants were given a survey in the 2016-2017 cohort and of that number 3,194 (91%) responded to the survey, respectively. We have added total number of the first-year students entering the University. The change has been made as suggested in Figure 1. See Page 3, Line 97-98 and Line 112. See Page 5, Figure 1.

Reviewer’s comment: 3. On the other hand, table 1 is cconfuse, you should put mean +/- SD and n (%). In addition, the percentage should be made on the top total, for example smokers would be 100 (44/8159) = 0.54% in male and in female 100 (7/4015) = 0.17%, which clearly shows the difference of smokers p = 0.003 in favor of males. The same for the other qualitative variables.

Response: We thank Reviewer 1 for the comments. The change has been made as suggested. See Table 1.

Reviewer 2 Report

To the authors, 

The study sample flowchart has numbers that don't add up. For example, if you substract 3,144 (Freshman who completed survey only once) from 12,174, you get 9,030 in the rectangle below the 12,174 rectangle. However, here they got 10,383 + 1,831 = 12,214.  The numbers don't match either in the rectangles below. Please fix this or provide an explanation to the reader.

Air Quality Index (AQI) is a 0-500 scale used for health warning purposes, so basically it does not units. Somehow here, you gave it a unit in ug/m3, unless you mean Ozone (O3). Please explain the reason behind using units for a scale. It is okay to use units (ug/m3) for PM2.5 and10, and NO3, but not for a scale unless you mean something else not commonly known in the field of air pollution.

How reliable is the self-reported SB (Sedentary Behavior)? Did any other study shed a light on this?

The objective measures (such as accelerometers) do exist these days. The same can said about the self-reported measures.

Please make sure your manuscript is reviewed by a native English speaker or someone with excellent command of English.       

Author Response

We want to thank you for your helpful comments, which have helped significantly improve the manuscript. We have summarized our point-by-point responses/revisions to the comments and revisions made based on your helpful comments! Again, thank you very much!

Reviewer #2:

Reviewer’s comment: 1. The study sample flowchart has numbers that don't add up. For example, if you substract 3,144 (Freshman who completed survey only once) from 12,174, you get 9,030 in the rectangle below the 12,174 rectangle. However, here they got 10,383 + 1,831 = 12,214.  The numbers don't match either in the rectangles below. Please fix this or provide an explanation to the reader.

Response:

We thank Reviewer 2 for the comment. We have taken the comments/suggestions very seriously and have made all double-check the flowchart numbers. We have added total number of the survey (13,500) and corrected our mistake the number 1,791 (Freshman who completed survey only once) and deleted the number 1,831 (Freshmen who responded to the survey more than 3 times), which was including in 10,383 (Freshmen who responded to the survey more than 2 times). We have fix the number. See Page 5, Figure 1. Again, we want to express our thankful to reviewer 2’s meaningful comments!

Reviewer’s comment: 2. Air Quality Index (AQI) is a 0-500 scale used for health warning purposes, so basically it does not units. Somehow here, you gave it a unit in ug/m3, unless you mean Ozone (O3). Please explain the reason behind using units for a scale. It is okay to use units (ug/m3) for PM2.5 and10, and NO3, but not for a scale unless you mean something else not commonly known in the field of air pollution.

Response: Thank you very much for the reviewer’s comments. It is true that (AQI) is a 0-500 scale based on the level of six atmospheric pollutants used for health warning purposes in China, it does not units. We removed the units of AQI. See Page4, Line 146, Page 6, Line 206-207 and Table 2.

Reviewer’s comment: 3. How reliable is the self-reported SB (Sedentary Behavior)? Did any other study shed a light on this? The objective measures (such as accelerometers) do exist these days. The same can said about the self-reported measures.

Response: Thanks for the comments. The reliability coefficients for the self-reported (IPAQ-SF) SB was 0.50-0.94 (12-Country research) [1] and was 0.97 (0.95—0.98) in Chinese adults [2]. It is ture that SB behavior was self-reported and subject to recall error and social desirability bias, participants usually may tend to report less sedentary behavior. Future research should use more objective ways to estimate sedentary behavior (e.g. accelerometers or activPAL). We have added this limitation in the discussion section. See Page 11, Line 487-789.

Reviewer’s comment: 4. Please make sure your manuscript is reviewed by a native English speaker or someone with excellent command of English. 

Response: Thanks for the comments. A native English speaker who is our coauthor Shelby Paige Gordon again has proofread the revised manuscript. The sections of the manuscript that were edited are colored in yellow. We believe the readability of this manuscript has improved as a result.

1.  Macfarlane, D. J.; Lee, C. C. Y.; Flo, E. Y.; Chan, K. L.; Chan, D. T. S., Reliability and validity of the Chinese version of IPAQ (short, last 7 days). J Sci Med Sport. 2007, 10, (1), 45-51.

2.    Craig, C. L.; Marshall, A. L.; Sjostrom, M.; Bauman, A. E.; Booth, M. L.; Ainsworth, B. E.; Pratt, M.; Ekelund, U.; Yngve, A.; Sallis, J. F.; Oja, P., International physical activity questionnaire: 12-country reliability and validity. Med Sci Sports Exerc. 2003, 35, (8), 1381-95.